# Analysis of the efficacy of splenic artery superselective embolization in cirrhosis with hepatocellular carcinoma

**Ke Zhao**, **Hong-Tao Hu\***, **Hai-Liang Li**, **Hong-Tao Cheng**, **Ya-nan Zhao**, **Yuan Hang**, **Quan-jun Yao**

Department of Minimal-Invasive Intervention, The Affiliated Cancer Hospital of Zhengzhou University & Henan Cancer Hospital, Zhengzhou City, Henan Province, China

\* huhongtaogy@163.com

## Abstract

### Background

To explore the safety and effectiveness of partial splenic embolization (PSE) in patients with hypersplenism and hepatocellular carcinoma (HCC) and to compare the efficacy of superselective and non-superselective embolization of splenic artery branches.

### Procedure

We retrospectively analyzed 64 patients with HCC who underwent PSE between August 2020 and December 2022. The patients were categorized into two groups based on different treatment plans: Group A (n=33) underwent superselective embolization and Group B (n=31) underwent non-superselective embolization of the splenic artery branches. The safety and effectiveness of the two methods were evaluated along with changes in peripheral blood cells [mainly white blood cells (WBC) and red blood cells (RBC)] and platelet (PLT) counts at different time points after PSE. Postoperative adverse events were also compared between the two groups,

### Results

The technical success rate was 100% for both procedures. The PLT and WBC counts of the two groups significantly increased one week after PSE ($P<0.05$), and there was no statistically significant difference in the RBC count changes. At follow-up (4, 16, and 24 weeks), the PLT and WBC counts remained consistent at levels which were significantly different from those before PSE ($P<0.05$). However, the RBC counts were not significantly different ($P>0.05$). An independent sample t-test was used to compare the differences in blood counts between the two groups at the same time point. There were no statistically significant differences in PLT, WBC, and RBC

**Data availability statement:** All relevant data are within the manuscript and its Supporting Information files.

**Funding:** The author(s) received no specific funding for this work.

**Competing interests:** The authors have declared that no competing interests exist.

counts between Group A and Group B at any time point after PSE (P>0.05). The incidence of fever and pain in Group B was significantly higher than that in Group A (P<0.05),

## Conclusion

Partial splenic artery embolization is a safe and effective treatment option for hypersplenism. Both splenic artery branch superselective and non-superselective embolization strategies demonstrated comparable outcomes. However, superselective embolization exhibited a lower incidence of postprocedural complications than non-superselective embolization.

## Introduction

Hepatitis B is prevalent in China. This viral infection can trigger hepatic inflammation and cell death, leading to the development of liver cirrhosis in a significant number of individuals, particularly in areas of high prevalence. Cirrhosis has been well recognized as a critical precursor to hepatocellular carcinoma (HCC), with over 85% of HCC cases exhibiting this condition [1]. Splenic hyperfunction emerges as a frequent comorbidity of liver cirrhosis, often resulting in splenic congestion, thrombocytopenia, leukopenia, and a spectrum of complications, including bleeding and infections [2,3]. For patients with HCC and underlying liver cirrhosis, transarterial chemoembolization (TACE) presents particular challenges owing to the potential for hemorrhagic complications stemming from the treatment or from portal hypertension. In particular, thrombocytopenia is associated with an increased risk of bleeding and mortality during and after TACE. Additionally, chemotherapeutic agents administered during TACE can intensify cytopenia by inducing myelosuppression [4–6].

Although splenic artery embolization has proven to be an effective treatment for hypersplenism in patients with cirrhosis and severe thrombocytopenia, limited research has been conducted on the feasibility, safety, and effectiveness in managing portal hypertension in patients with HCC [7]. Kim et al. recently reported that partial splenic embolization (PSE) is safe and effective for increasing platelet counts in patients with HCC and thrombocytopenia undergoing TACE [8]. For HCC patients undergoing TACE for thrombocytopenia caused by hysplenia, PSE is an essential adjuvant treatment to improve platelet levels and ensure the safety and efficacy of TACE [8].

However, regarding safety, 80% of patients experience common (mild to moderate) complications such as postoperative fever and abdominal pain after PSE, as well as serious complications such as splenic abscess, pleural ascites, spontaneous bacterial peritonitis, and pancreatitis, which can lead to prolonged hospitalization. Ischemic necrosis is the main cause of post-embolization complications. We hypothesized that using splenic artery superselective embolization may reduce the extent of necrotic tissue resulting from ischemia, with the expectation of fewer clinical complications. Therefore, this study aimed to compare the efficacy of superselective and non-superselective splenic artery embolization in the treatment of hypersplenism in patients with cirrhotic HCC.

## Materials and methods

### Baseline data

We retrospectively analyzed patients who underwent splenic artery embolization at our center between August 2020 and December 2022. HCC was diagnosed according to the recommendations of the European Association for the Study of the Liver and the American Association for the Study of the Liver guidelines. Patients with vascular invasion or other related diagnoses from cancer were excluded from the study. Patients unsuitable for surgical resection were also excluded. Liver cirrhosis was diagnosed based on the "Chinese consensus on the management of liver cirrhosis" document (Chinese Society of Gastroenterology, Chinese Medical Association, 2023) [9]. The diagnosis of liver cirrhosis and hypersplenism was determined using computed tomography (CT). Key features included the following: (1) obvious widening of the hepatic fissure, wavy features of the hepatic margin, displaced gallbladder, and an enlarged gallbladder fossa, accompanied by imaging features such as uneven density of the liver parenchyma, thickened portal vein, collateral circulation, splenomegaly, and portal hypertension. (2) Splenomegaly was diagnosed when the maximum splenic diameter (maximum anteroposterior size of the splenic hilum) exceeded 12 cm in the transverse CT plane. The exclusion criteria were as follows: (1) patients who could not be followed up consistently after surgery and (2) patients with other blood disorders causing platelet reduction. The splenic infarction rate was defined as the ratio of the infarcted spleen volume to the pre-infarction spleen volume. The infarcted spleen volume was calculated as the difference in these volumes prior to and after PSE. All patients survived for more than 6 months postoperatively.

This retrospective study was approved by the Ethics Committee of Henan Province Cancer Hospital (Approval No. 2016ct005) who waived the requirement for informed consent. The data used in this study were obtained from an in-house clinical database.

Patients were categorized into Group A and Group B according to the different methods of splenic artery embolization. Group A underwent superselective splenic artery branch embolization, and Group B underwent conventional PSE. Based on the high level of pain difference between the groups (35%) and a significance level of 0.05, with a statistical efficacy of 80%, our analysis indicated that approximately 27 patients were required in each group.

### Splenic artery embolization

All patients signed a consent form prior to surgery. Patients were placed in a supine position. Using the modified Seldinger technique, the right femoral artery was punctured under local anesthesia and a 5F sheath was inserted. Under the guidance and monitoring of digital subtraction angiography (DSA) (GE Company), a 5F RH (Terumo Medical Products Co., LTD) catheter was selectively inserted into the abdominal aorta, and 16 ml of iodine-containing contrast agent was injected at a rate of 4 ml/s to show the course and distribution of the splenic artery (Fig 1A, D). For Group A embolization, a microcatheter (HENGRUI MEDICAL) was used for "super-selection" of the splenic artery branch through the 5F RH catheter (Fig 1B); 710–1000 micron gelatin sponge (Hangzhou Alicon Pharm SCI (TEC CO., LTD) particles mixed with an appropriate amount of contrast agent and gentamicin (16 mg) were injected for embolization (Fig 1C). For Group B embolization, based on the tortuosity of the splenic artery, the RH catheter or microcatheter was inserted into the main splenic artery, avoiding the pancreatic and dorsal pancreatic artery, and 710–1000 micron gelatin sponge particles were mixed with an appropriate amount of contrast agent. Gentamicin (16 mg) was then injected for embolization (Fig 1E). Follow-up angiography was performed after embolization. Based on the changes in the splenic artery blood flow velocity and branch imaging after embolization, C-arm CT scans were performed to control the amount of splenic embolism to approximately 30%–50%.

### Evaluation indicators

Safety evaluation indicators included: (1) Intraoperative complications: embolization process leading to the formation of dissection or splenic artery aneurysm and instrument fracture in the blood vessel during the operation. (2) Postoperative

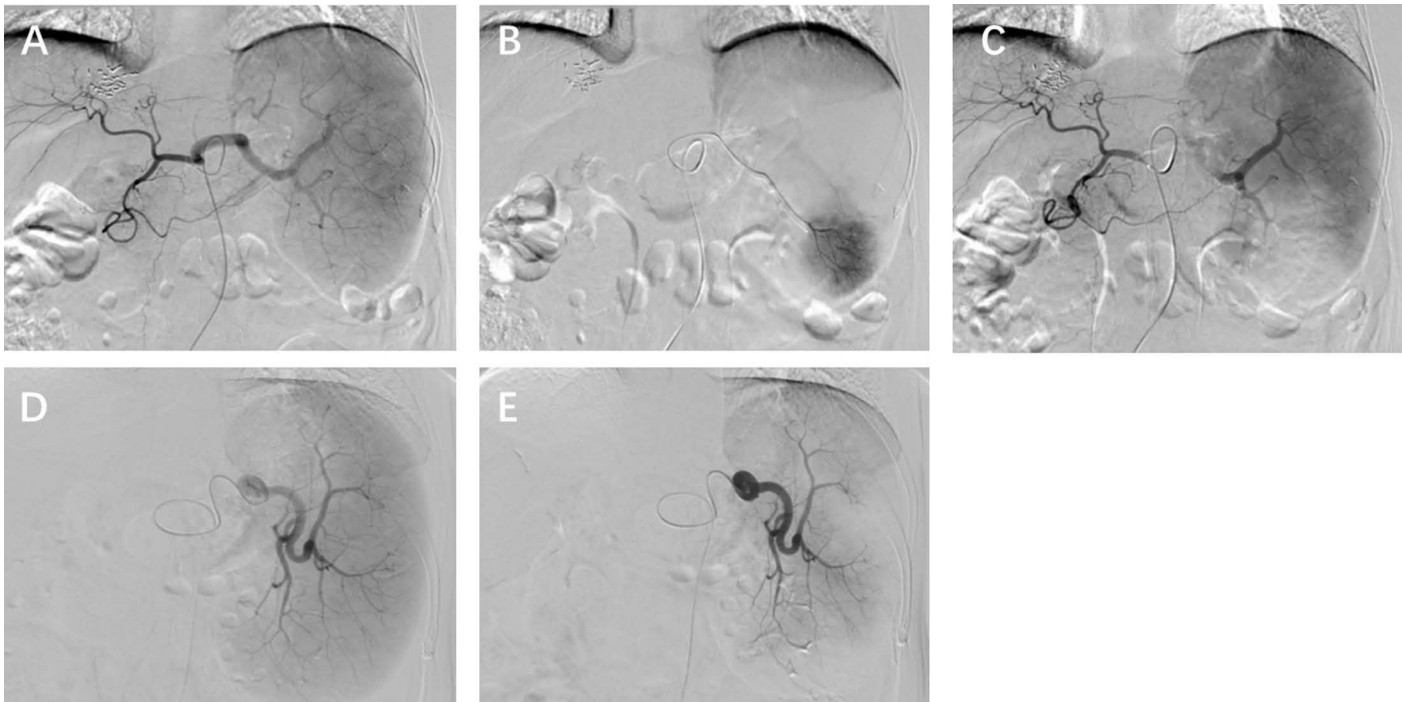

**Fig 1. Splenogram of each group of patients.** A. D. Under the guidance and monitoring of digital subtraction angiography, a 5F RH catheter was selectively inserted into the abdominal aorta, and a total of 16 ml of iodine-containing contrast agent was injected at a rate of 4 ml/s to show the course and distribution of the splenic artery. B. Figure shows the inferior pole branch of the splenic artery which was "super-selected" through a 5F RH catheter using a microcatheter and then imaged. C. Gelfoam particles of 710-1000 μm were mixed with an appropriate amount of contrast medium and gentami-cin (16 mg) and injected through a microcatheter for embolization. After embolization, approximately 1/3 of the staining in the inferior pole of the spleen disappeared. E. Post-embolization angiography [with 710-1000 μm gelatin sponge particles injected through an RH catheter or microcatheter and mixed with the appropriate amount of contrast medium and gentamicin (16 mg)] shows disappearance of peripheral staining in the spleen.

complications: mild complications such as fever, pain, nausea, and vomiting, and moderate-to-severe complications such as pleural effusion/ascites, splenic abscess, and pancreatitis. Effectiveness evaluation indicators included white blood cell and platelet counts which were evaluated before and at 1, 4, 12, and 24 weeks after PSE.

The extent of splenic infarction after PSE can be estimated by CT or magnetic resonance imaging (MRI) [10]. A simplified calculation was used to measure the volume of the infarct area in the postoperative image. This value was then compared with the total volume of the spleen to calculate the infarction rate, expressed as the percentage of the infarct volume to the total spleen. The Numeric Rating Scale (NRS) is a tool frequently used to assess pain intensity and perception in the clinic and for research purposes [10]. The NRS scores range from 0 to 10 where 0 represents "no pain" and 10 represents "worst pain imaginable".

## Statistical methods

Continuous variables are expressed as mean ± standard deviation (SD) or interquartile [median (Q1, Q3)] according to the normality of the data, and the two groups were compared using an independent sample t-test or Mann-Whitney U test. Enumeration data are expressed as percentages (%) which were compared between the groups using the chi-square test. As PLT, RBC, and WBC were included in the repeated measurement data analysis to analyze the efficacy in postoperative patients, the Friedman test or one-way repeated measures analysis of variance (ANOVA) test was used according to the

normality of data. Statistical analyses were performed using the R software version 3.6.1 (Foundation for Statistical Computing, Vienna, Austria). Statistical significance was set at $P < 0.05$.

## Results

### Patient baseline information and feasibility

A total of 64 patients with complete clinical and follow-up data were selected, including 46 males (71.9%) and 18 females (28.1%), with an average age of 55.2 years. Among them, 33 underwent superselective embolization of the splenic artery branches (Group A), and 31 underwent non-superselective embolization of the splenic artery branches (Group B). In Group A, 21 patients (63.6%) had Child-Pugh class A liver function, and 12 patients (36.4%) had Child-Pugh class B; in Group B, 18 patients (58.1%) had Child-Pugh class A liver function, and 13 patients (41.9%) had Child-Pugh class B. Blood test indices, including PLT, WBC, and RBC counts, decreased by varying amounts in both groups. There were no significant differences in the counts between the two groups ($P > 0.05$). The clinical data of the two groups are presented in Table 1.

All patients underwent successful femoral artery puncture, sheath insertion, and superselective catheterization. Arterial angiography performed after successful catheterization showed a tortuous and enlarged splenic artery that was effectively embolized using gelatin sponge particles through a catheter or microcatheter. Post-embolization angiography showed no significant visualization of the splenic artery branches or parts of the parenchyma. Postoperative CT scans exhibited embolization volumes ranging from 30% to 50%. The technical achievement rate was 100% in both groups.

### Safety

No patient experienced major procedure-related complications. Fever, pain, nausea/vomiting, hydrothorax/ascites, splenic impotence, and pancreatitis occurred in 18 (28.1%), 61 (95.3%), 51 (79.7%), 12 (18.7%), 0 (0.0%), and two

Table 1. Demographics and clinical characteristics of patients undergoing treatment.

|  | Group A (n = 33) | Group B (n = 31) | All (n = 64) | P value |
|---|---|---|---|---|
| Sex |  |  |  | 0.689 |
| Male | 23 (69.7%) | 23 (74.2%) | 46 (71.9%) |  |
| Female | 10 (30.3%) | 8 (25.8%) | 18 (28.1%) |  |
| Age (min-max) | 55.758 (35-73) | 54.677 (43-64) | 55.234 (35-73) | 0.595 |
| Child-Pugh |  |  |  | 0.557 |
| A | 21 (63.6%) | 18 (58.1%) | 39 (60.9%) |  |
| B | 12 (36.4%) | 13 (41.9%) | 25 (39.1%) |  |
| Hypertension | 5 (15.2%) | 10 (32.3%) | 15 (23.4%) | 0.106 |
| Diabetes | 4 (12.1%) | 3 (9.7%) | 7 (10.9%) | 0.754 |
| Etiology-HBV | 22 (66.7%) | 23 (742%) | 45 (70.3%) | 0.510 |
| WBC | 2.71 ± 1.10 | 3.11 ± 1.25 |  | 0.183 |
| RBC | 3.64 ± 0.72 | 3.77 ± 0.87 |  | 0.504 |
| PLT | 45.0 (33.0, 62.0) | 51.0 (34.0, 65.0) |  | 0.519 |
| ALT | 26.0 (20.5, 35.5) | 20.0 (13.0, 31.0) |  | 0.066 |
| AST | 36.0 (29.0, 45.0) | 28.0 (21.0, 49.0) |  | 0.150 |
| ALB | 35.24 ± 4.79 | 36.53 ± 5.87 |  | 0.338 |

Note: HBV: Hepatitis B Virus; WBC: White Blood Cell; RBC: Red Blood Cell; PLT: Platelet; ALT: Alanine Aminotransferase; AST: Aspartate Aminotransferase; ALB: Albumin.

patients (3.1%), respectively. There was no significant difference in the incidence of fever or pain between the two groups (P > 0.05).

The incidence of nausea and/or vomiting in Group A was significantly lower than that in Group B (66.7% vs. 93.5%, P = 0.008). The incidence of pleural effusion and ascites in Group A was also lower than that in Group B (9.1% vs. 29.0%, P = 0.041). No splenic abscesses were observed in either group. Pancreatitis occurred in two patients in Group B. Details of the postoperative complications are shown in Table 2. All complications were treated symptomatically, and all patients recovered.

The difference in pain at different degrees between the two groups was further analyzed using the NRS. Twenty-one patients (63.6%) in Group A scored 0–3 on the NRS, indicating mild pain. Twenty-two patients (71.0%) in Group B scored ≥4, indicating moderate to severe pain. There was a statistically significant difference in pain between the two groups (Table 3). For mild–moderate pain, patients received non-steroidal anti-inflammatories (NSAIDs) or acetaminophen, whereas those with moderate–severe pain (NRS ≥ 4) were given short-term opioids (e.g., oxycodone). Most patients reported significant relief within 48–72 hours.

**Table 2. Incidence of complications between groups (except pain).**

|  | Group A (n = 33) | Group B (n = 31) | All (n = 64) | P value |
|---|---|---|---|---|
| Fever |  |  |  | 0.876 |
| yes | 9 (27.3%) | 9 (29.0%) | 18 (28.1%) |  |
| no | 24 (72.7%) | 22 (71.0%) | 46 (71.9%) |  |
| Pain |  |  |  | 0.259 |
| yes | 30 (90.9%) | 31 (100.0%) | 61 (95.3%) |  |
| no | 3 (9.1%) | 0 (0.0%) | 3 (4.7%) |  |
| Nausea/Vomit |  |  |  | 0.008 |
| yes | 22 (66.7%) | 29 (93.5%) | 51 (79.7%) |  |
| no | 11 (33.3%) | 2 (6.5%) | 13 (20.3%) |  |
| Hydrothorax/Ascites |  |  |  | 0.041 |
| yes | 3 (9.1%) | 9 (29%) | 12 (18.7%) |  |
| no | 30 (90.9%) | 22 (71%) | 52 (81.3%) |  |
| Splenic abscess |  |  |  | 1.000 |
| Yes | 0 (0.0%) | 0 (0.0%) | 0 (0.0%) |  |
| no | 33 (100.0%) | 31 (100.0%) | 64 (100.0%) |  |
| Pancreatitis |  |  |  |  |
| yes | 0 (0.0%) | 2 (6.5%) | 2 (3.1%) | 0.445 |
| no | 33 (100.0%) | 29 (93.5%) | 62 (96.9%) |  |

**Table 3. Pain score for Groups A and B.**

| Numeric pain scale (NRS) | A group | B group | All | P value |
|---|---|---|---|---|
| 0 | 3 | 0 | 3 | 0.022 |
| 1-3 | 18 | 9 | 27 |  |
| 4-6 | 9 | 15 | 24 |  |
| 7-9 | 3 | 7 | 10 |  |
| 10 | 0 | 0 | 0 |  |

### Changes of blood indices after the different surgical methods

**Pair-wise comparison preoperatively and one week postoperatively.** The WBC, RBC, and PLT counts in all cohorts tended to increase in the first week after surgery and then remained at a certain level. Table 4 shows the mean count and variation in PLT, WBC, and RBC counts at weeks 1, 4, 12, and 24 in Groups A and B. An independent sample t-test was used to assess the differences between the different blood parameters during the same period between Groups A and B. The results are shown in Table 4. There were no statistically significant differences in PLT, WBC, and RBC counts between the two groups at 1, 4, 12, and 24 weeks after PSE, indicating no difference in the therapeutic effect between the two groups.

**Effect of different surgical methods on blood parameters at 1, 4, 12, and 24 weeks.** Finally, to evaluate the statistical differences between the PLT, WBC, and RBC counts at 1, 4, 12, and 24 weeks after surgery in the groups, the Friedman test or one-way repeated measures ANOVA test was used, and a correlation graph was drawn (Fig 2). The PLT and WBC counts at 1, 4, 12, and 24 weeks after surgery were significantly higher than those before surgery in both groups. Additionally, there was no difference between the different time points after surgery, indicating that the PLT and WBC counts were elevated and maintained at a certain level postoperatively. There was no difference in the WBC count at any stage (including the preoperative stage).

## Discussion

Our study showed that both superselective and non-superselective embolization of the splenic artery branches resulted in significant improvements in peripheral WBC and PLT counts after embolization. These improvements were observed at 1, 4, 12, and 24 weeks after embolization, with statistically significant increases ($P < 0.05$) compared with the pre-embolization levels. Moreover, the improvements were relatively maintained after the procedure. Notably, there were no significant differences in PLT and WBC counts between the two groups at different time points. This suggests that using superselective splenic artery embolization for hypersplenism in patients with cirrhosis and HCC is equivalent to using non-superselective splenic artery embolization. In addition, the incidence of postoperative pain, hydrothorax, and ascites in Group A was significantly lower than that in Group B ($P < 0.05$), suggesting that superselective splenic artery embolization was safer. These results indicate that superselective PSE is effective and may be safer for the treatment of such patients.

**Table 4. Comparison of WBC, RBC, and PLT between Groups A and B at the same follow-up time point.**

| Follow-up time | Indicators | A group | B group | P value |
|---|---|---|---|---|
| 1week | PLT | 67.0 (57.0, 105.5) | 85.0 (65.0, 127.0) | 0.094 |
| | WBC | 6.02±2.41 | 6.82±2.72 | 0.222 |
| | RBC | 43.66±0.65 | 3.82±0.73 | 0.371 |
| 4week | PLT | 80.0 (59.5, 119.5) | 88.0 (68.0, 128.0) | 0.354 |
| | WBC | 4.85±2.29 | 5.11±2.42 | 0.665 |
| | RBC | 3.65±0.85 | 3.76±0.83 | 0.602 |
| 12week | PLT | 87.0 (59.0, 135.0) | 88.0 (66.0, 149.0) | 0.605 |
| | WBC | 4.75±1.87 | 5.53±2.51 | 0.165 |
| | RBC | 3.58±0.75 | 3.75±0.65 | 0.316 |
| 24week | PLT | 82.0 (51.0, 121.0) | 91.0 (59.0, 125.0) | 0.519 |
| | WBC | 5.48±2.18 | 6.03±1.83 | 0.281 |
| | RBC | 3.52±0.84 | 3.80±0.75 | 0.168 |

Note: WBC: White Blood Cell; RBC: Red Blood Cell; PLT: Platelet.

Patients with liver cirrhosis often show symptoms of portal hypertension, leading to the obstruction of splenic venous return and increased pressure on the spleen. These phenomena result in splenomegaly and splenic dysfunction (hypersplenism) [11]. Hyperplenism refers to the decrease in one or more types of blood cells and splenomegaly in the presence of normal hematopoietic function in the bone marrow. When the spleen becomes hyperactive, the mononuclear macrophage system proliferates and becomes more active, resulting in the excessive destruction of blood cells and a decrease in all three cell lines, which can lead to complications such as infection, anemia, and bleeding. Splenic artery embolization can reduce splenic blood flow, improve blood flow stasis, reduce portal pressure, and weaken splenic function, thereby decreasing the risk of gastrointestinal bleeding and improving low WBC and platelet levels in patients [12]. The results of this study indicate that both superselective and non-superselective embolization of splenic artery branches are feasible options for clinical treatment. The WBC and PLT counts of Groups A and B were significantly increased in the first week after surgery and at several different time points (1 month, 3 months, and 6 months) during dynamic follow-up (P<0.05), consistent with previous studies. However, there were no significant differences in the WBC and PLT counts between the two groups at the same time points after embolization (P>0.05), indicating that both types of embolization were effective.

Post-embolization syndrome, characterized by fever and pain, is commonly observed after PSE. Research has shown that the duration of fever is associated with the extent and degree of splenic parenchymal infarction. Prompt anti-inflammatory treatment is essential to prevent the development of splenic abscesses [13]. Tissue ischemia, infarction, liquefaction absorption, inflammation, edema, and exudation may occur after splenic artery embolization (SAE) [14]. These cause increased tension in the splenic capsule which irritates the diaphragm, pleura, and peritoneum, leading to varying degrees of pain in most patients. Our results showed no statistically significant difference in pain between patients undergoing superselective and those undergoing non-superselective embolization of the splenic artery branches (P>0.05). However, the NRS scores indicated significantly lower pain levels in Group A than Group B (P<0.05). Zhu et al. found that the incidence of complications in embolization patients increased with the embolization volume, and Child-Pugh class C and large splenic infarction volume were identified as independent risk factors for PSE complications [13,14]. In the present study, the embolization volumes were all less than 50% and no patients had Child-Pugh class C liver function; therefore, no severe complications were observed. Kim et al. reported a 100% incidence of post-embolization syndrome; however, no complications were severe. However, Kim et al.'s study involved embolization of 70%–80% of the spleen, a higher proportion than that reported in other research, warranting further investigation [8]. We limited the infarction to 30–50% of the splenic volume; fever generally lasted less than 3–4 days, and no splenic abscesses occurred.

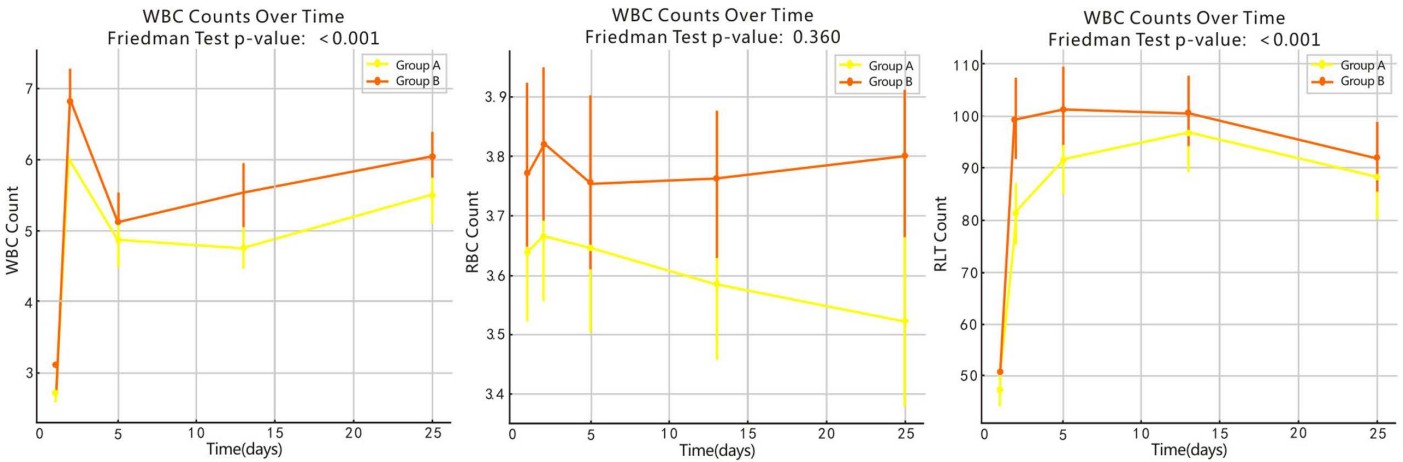

**Fig 2. Visual Plot of Friedman Test or One-Way Repeated Measures ANOVA Test.**

For operative embolic materials, we chose 710–1000 µm gelatin sponge particles to achieve predictable partial infarction (approximately 30%–50% of splenic volume) by effectively occluding small-to-medium branches while minimizing penetration into very small vessels. Compared with smaller (<300–500 µm) or larger (>1100 µm) particles, this size range better balances efficacy and safety[15]. Unlike permanent materials such as polyvinyl alcohol (PVA) or metallic coils, gelatin sponges degrade over time, reducing the risk of extensive necrosis or abscess formation.

Nausea and vomiting are common complications of PSE. The incidence of nausea and vomiting was significantly lower in Group A than Group B (P<0.05), which may be related to the post-embolization redistribution of blood flow [16]. Similarly, the incidence of pleural effusion and ascites in Group A was significantly lower than that in Group B, potentially owing to differences in embolization sites [17]. In Group A, embolization of the lower pole of the splenic artery was superselective, resulting in less irritation of the diaphragm and pleura. Pancreatitis and splenic abscess are the most severe complications associated with partial splenic artery embolization. Pancreatitis may also be associated with pancreatic artery embolization. Our study revealed no cases of pancreatitis in Group A whereas two cases were observed in Group B. Superselective microcatheter techniques were used to cannulate the splenic artery branches in Group A, whereas embolic agents were directly injected into the main trunk of the splenic artery without further superselective procedures in Group B. The pancreatitis cases in Group B may have been caused by reflux of the embolic agent and embolization of the pancreatic arteries.

In addition to increasing platelet counts [16], PSE can improve liver function reserves, portal hypertension, and gastroesophageal varices. Reports indicate that performing PSE and TACE simultaneously can enhance the liver function reserves [18,19]. Ishikawa et al. demonstrated that reduced splenic blood flow along with increased hepatic and superior mesenteric artery blood flow improves liver function reserves [19]. For patients with HCC and thrombocytopenia, combining TACE and PSE may help maintain liver function reserves and potentially improve prognosis. The present study showed that superselective PSE was effective and had a lower incidence of adverse reactions, suggesting a more favorable role in the comprehensive treatment of HCC.

The limitations of this study include its retrospective and single-center design, small sample size, and lack of multicenter prospective studies. Some research has indicated that liver function may be affected following SAE. However, all patients in this study had liver cancer, which affects liver function, and many underwent PSE at some point during TACE. To reduce the risk of postoperative complications, no patients underwent simultaneous TACE and PSE. However, this approach may require multiple procedures and potentially delay TACE treatment. If TACE and PSE are performed simultaneously as a single multisite procedure, particularly with superselective PSE, adverse reactions such as pain could be effectively reduced. This is an area for future research focus.

In conclusion, regardless of whether selective embolization of the splenic artery branches was performed, SAE demonstrated clear short- and long-term clinical efficacy. While the efficacy of the two methods was not significantly different, superselective embolization of the splenic artery branches was associated with fewer complications than non-superselective embolization. Therefore, when performing SAE, if vascular conditions permit, super-selective catheterization of the lower pole branches of the splenic artery should be prioritized before embolization. If conditions do not allow this, non-superselective embolization can also achieve the desired effect, albeit with a higher likelihood of complications.

## Supporting information

**S1 File. Supplementary material (raw data of patients).**
(DOCX)

## Author contributions

**Conceptualization:** Hong-Tao Hu.

**Data curation:** Hong-Tao Hu, Hong-Tao Cheng.

**Funding acquisition:** Hong-Tao Hu.

**Investigation:** Ke Zhao, Yuan Hang, Quan-jun Yao.

**Methodology:** Ke Zhao.

**Resources:** Yuan Hang.

**Validation:** Hai-Liang Li.

**Visualization:** Quan-jun Yao.

**Writing – original draft:** Hai-Liang Li, Hong-Tao Cheng.

**Writing – review & editing:** Ya-nan Zhao.

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
