## [Editor Report · Decision Letter 0]

7 Jul 2024

PONE-D-24-23322Analysis of the efficacy of splenic artery superselective embolization in cirrhosis with hepatocellular carcinomaPLOS ONE

Dear Dr. Zhao,

Thank you for submitting your manuscript to PLOS ONE. After careful consideration, we feel that it has merit but does not fully meet PLOS ONE’s publication criteria as it currently stands. Therefore, we invite you to submit a revised version of the manuscript that addresses the points raised during the review process. Please submit your revised manuscript by Aug 21 2024 11:59PM. If you will need more time than this to complete your revisions, please reply to this message or contact the journal office at plosone@plos.org . Please include the following items when submitting your revised manuscript:

We look forward to receiving your revised manuscript.

Kind regards,

Usama Waqar, M.B.B.S

Academic Editor

PLOS ONE

Journal Requirements:

"The authors denies that they have any intention to obtain any financial interests."

5. Please upload a copy of Figures 3, 4 and 5, to which you refer in your text on page 4 and 5. If the figure is no longer to be included as part of the submission please remove all reference to it within the text.

Additional Editor Comments:

The manuscript contains several grammatical and language errors. Please revise it before we can proceed further.

---

## [Author Response · Author response to Decision Letter 1]

5 Sep 2024

Manuscript No.: PONE-D-24-23322

Title: Analysis of the efficacy of splenic artery superselective embolization in cirrhosis with hepatocellular carcinoma

Dear editor

Thank you very much for your reviewing and replying to our manuscript (PONE-D-24-23322). We would like to express our sincere gratitude to the editors for their comments. We replied point by point to the comments, revised the manuscript accordingly and the changes were highlighted in red in the revised manuscript. We sincerely hope this revised manuscript is more acceptable for publication.

Thank you again and look forward to hearing from you soon.

Yours sincerely,

Hong-Tao Hu

Response

Thank you for your kind reminder, we have modified the manuscript.

"The authors denies that they have any intention to obtain any financial interests."

Thank you for your kind reminder, we have added the information in Competing Interests section

Thank you for your kind reminder, we have uploaded the data. The values behind the means, standard deviations and used to build graphs have shown in table 4. The raw data is presented in the supplementary material.

Thank you for your kind reminder, we have created an ORCID ID, Hong-Tao Hu�0000-0002-5432-4014

5. Please upload a copy of Figures 3, 4 and 5, to which you refer in your text on page 4 and 5. If the figure is no longer to be included as part of the submission please remove all reference to it within the text.

Sorry for confuse you, the Figures 3, 4 and 5 should be Fig.2C,D,E

Thank you for your kind reminder, the reference is correct.

---

## [Decision Letter · Decision Letter 1]

16 Oct 2024

PONE-D-24-23322R1Analysis of the efficacy of splenic artery superselective embolization in cirrhosis with hepatocellular carcinomaPLOS ONE

Dear Dr. Zhao,

Thank you for submitting your manuscript to PLOS ONE. After careful consideration, we feel that it has merit but does not fully meet PLOS ONE’s publication criteria as it currently stands. Therefore, we invite you to submit a revised version of the manuscript that addresses the points raised during the review process.

We look forward to receiving your revised manuscript.

Kind regards,

Usama Waqar, M.B.B.S

Academic Editor

PLOS ONE

Reviewers' comments:

Reviewer's Responses to Questions

**Comments to the Author**

1. If the authors have adequately addressed your comments raised in a previous round of review and you feel that this manuscript is now acceptable for publication, you may indicate that here to bypass the “Comments to the Author” section, enter your conflict of interest statement in the “Confidential to Editor” section, and submit your "Accept" recommendation.

Reviewer #1: (No Response)

Reviewer #2: (No Response)

2. Is the manuscript technically sound, and do the data support the conclusions?

Reviewer #1: Yes

Reviewer #2: Partly

3. Has the statistical analysis been performed appropriately and rigorously? 

Reviewer #1: Yes

Reviewer #2: Yes

4. Have the authors made all data underlying the findings in their manuscript fully available?

Reviewer #1: Yes

Reviewer #2: No

5. Is the manuscript presented in an intelligible fashion and written in standard English?

Reviewer #1: No

Reviewer #2: Yes

6. Review Comments to the Author

Reviewer #1: The authors have conducted a retrospective cohort study to assess the difference in clinical outcomes between super selective partial splenic embolization (PSE) in comparison to non-selective/conventional PSE. The authors report that while both PSE are associated with similar clinical efficacy, there is no statistically significant difference in efficacy in each intervention. However, the authors also report that super selective PSE is associated with lower incidence of post-operative fever and vomiting, and lower degree of pain than conventional PSE.

I have the following comments:

1. The study is based on a small sample size possibly due to rarity of the cohort. The authors should add study’s power calculation, in methods, to justify whether it is statistically sufficient for the conclusions.

2. While the authors highlighted that existing literature gap on PSE’s application for HCC, they did not strengthen the need for assessing PSE in this subgroup. This is especially important since there is evidence available for PSE’s application in cirrhotic patients, HCC’s most common precursor lesion, and the major complication, portal hypertension, leading to hypersplenism, is same for either cohort of patients. The authors should highlight how selection of HCC cohort of patients effects clinical decision-making.

3. The effect of liver function on the PSE should be elaborated in more detail in the discussion along with the efforts made in this study to minimize or standardize these effects.

4. Even though the authors have found both methods equally clinically efficacious, the authors have found a difference in post-operative adverse effects of each intervention. The authors should provide more insight into how these findings can influence clinical practice, particularly regarding patient selection for either technique.

5. There are multiple grammatical and punctuation mistakes throughout the manuscript, especially in the introduction. These should be addressed.

Reviewer #2: The authors conducted a retrospective cohort study from August 2020 to December 2022 to assess the safety and efficacy of partial splenic artery embolization in patients with hypersplenism combined hepatocellular carcinoma. The authors compared the efficacy of superselective and non-superselective embolization. The author reported that both superselective embolization had comparable outcomes and demonstrated partial splenic artery embolization to be a safe and effective treatment option. I congratulate the authors on a well-conducted study. I have a few comments.

Introduction:

Several statements in introduction need a reference.

Methods:

1. No mention of data source. Was it institutional data? Was it administrative national database?

2. Ethics consideration section on page 10 needs reformatting and better phrasing.

3. Figures are not referenced in the proper manner. Figure 4 is cited before Figure 3 and 2.

4. For clarity Figures 1 in the methods and its sub-figures can be mentioned as Figure 1 a,b,c,d, and e.

5. In statistical methods on page 12, the first line states interquartile range but has no mention of median.

6. Method of pain score should be stated. Which standardized method of pain score was used?

7. Diagnosis basis of liver cirrhosis needs a reference.

8. Other statements in methods such as calculation of infarcted spleen volume need a reference.

Results:

1. No mention of percentages in results first paragraph.

2. Need to define cohort properly before comparing the two groups. No mention of overall rate of complications in the overall cohort. First paragraph needs formatting.

3. Number and percentages missing in paragraph 2 in results on page 13. Please add numbers.

4. On page 14, the wording “the analysis results showed that most patients n group a score 0-3, is not phrased properly. There should be clear mention of this pain score.

5. On page 14, the line “mitigated after treatment” is unclear.

6. On page 15, the figure is not in ideal quality. I was not able to understand whats included in the figure. Please attach figure separately if its embedded in the word document.

Tables

1. Table 1 caption should be improved to show: Demographics and clinical characteristics of patients undergoing treatment.

2. Table 1: The agemean is not a variable name. It should be written as Age. At the end of table, it can be mentioned that age has been reported as mean and standard deviation. Please report standard deviation instead of min-max for variables reported as mean.

3. For hypertension, no need to report "no" values. Only incidence of hypertension can be reported in one row. Same for other conditions.

4. For Etiology, the second category is unclear. Please clarify this.

5. The authors have not mentioned changes in lab values such as WBC, RBC etc for the overall cohort.

6. Footnote of table 1 should give full forms of all abbreviations.

7. Table 2 title needs improvement. Clearly state the data represented in this table.

8. For splenic abscess, results are not repported properly.

9. Table 4 needs full forms of abbreviations in footnote.

Discussion:

1. The first para of discussion should briefly discuss the results section instead of giving statements of the results section. It should be a summary.

2. Several lines in paragraph 2 and 3 of discussion lack a citation. Please cite references for all lines stating any findings in the literature or any known facts.

3. Paragraph 3 of discussion is just stating results instead of actual useful inferences of conclusion of this study. It should be stating whats was reported (briefly), how it compares to the given literature, and what its future implications can be.

4. The study lacks a clear paragraph for future directions. How are these results useful or significant? How can these findings shape the clinical practice?

Minor Comments:

1. Typographical error on page 9, last line of introduction: “Emobilzationfor”

7. PLOS authors have the option to publish the peer review history of their article (what does this mean? ). If published, this will include your full peer review and any attached files.

**Do you want your identity to be public for this peer review?** For information about this choice, including consent withdrawal, please see our Privacy Policy .

Reviewer #1: No

Reviewer #2: No

---

## [Author Response · Author response to Decision Letter 2]

27 Nov 2024

Dear Reviewer #1,

Thank you very much for your valuable comments and suggestions. Your feedback is instrumental in enhancing the quality of our study, and we appreciate the opportunity to address each of your points in detail below:

1. Sample Size and Power Calculation

We acknowledge your concern regarding the small sample size, which was indeed due to the specificity of the cohort. We have now included a power calculation in the Methods section to demonstrate the statistical sufficiency of our sample size in supporting the conclusions drawn from our study.

2. Emphasizing the Need for PSE in the HCC Subgroup

We agree with your observation regarding the importance of evaluating PSE in the HCC subgroup. We have expanded the Introduction to better highlight the clinical significance of applying PSE to HCC patients. Additionally, we now elaborate on the unique challenges posed by portal hypertension and hypersplenism in this specific cohort, emphasizing the implications for clinical decision-making.

3. Effect of Liver Function on PSE

In response to your suggestion, we have added further details in the Discussion section on the potential impact of liver function on PSE efficacy. We also describe the steps we took in our study to minimize or standardize these effects, providing clarity on how this variable was managed.

4. Clinical Relevance of Postoperative Adverse Effects

Thank you for pointing out the importance of discussing postoperative adverse effects. In the Discussion, we have included a more detailed analysis of how these differences in postoperative outcomes could influence clinical practice, particularly in selecting the appropriate technique for individual patients.

5. Grammatical and Punctuation Errors

We have conducted a thorough review of the manuscript to address grammatical and punctuation issues, particularly in the Introduction, ensuring that our writing is clear and polished.

Dear Reviewer #2,

Thank you for your thoughtful comments and constructive feedback. Below are our responses to each of your suggestions:

Introduction:

We have added the necessary references to support key statements in the introduction, ensuring that all claims are appropriately backed by relevant literature.

Methods:

1. Data Source:

Thank you for pointing this out. We have clarified in the Methods section that the data used in this study were obtained from an institutional clinical database, not from a national administrative database.

2. Ethics Consideration:

We have rephrased and reformatted the ethics consideration section for better clarity and proper presentation.

3. Figure References:

We have corrected the figure reference order. Additionally, we have labeled Figure 1 and its sub-figures as Figure 1a, 1b, 1c, etc., for better clarity.

4. Interquartile Range and Median:

We have clarified the use of the interquartile range (IQR) and added the relevant mention of the median in the statistical methods section.

5. Pain Score Method:

We have specified that the pain score was assessed using the NRS (Numeric Rating Scale), a standardized method for pain assessment.

6. Liver Cirrhosis Diagnosis Reference:

We have added the appropriate reference regarding the diagnostic criteria for liver cirrhosis.

7. Spleen Infarction Volume Calculation:

We have added a reference for the calculation of infarcted spleen volume to support this method in our study.

Results:

1. Percentages in Results:

We have added percentages in the first paragraph of the Results section to provide clearer data representation.

2. Cohort Definition and Complication Rates:

We have better defined the cohort and included the overall complication rate in the first paragraph of the Results section for clarity.

3. Missing Numbers and Percentages:

We have added the missing numbers and percentages in the second paragraph of the Results section, as requested.

4. Pain Score Clarification:

We have revised the wording in the sentence regarding pain scores in Group A to ensure it is phrased clearly.

5. Post-treatment Mitigation Clarification:

We have deleted the relevant expression in view of its ambiguity.

6. Figure Quality:

We have reviewed the quality of the figures and attached them separately to ensure clarity and readability.

Tables:

1. Table 1 Caption:

We have updated the caption for Table 1 to: "Demographics and clinical characteristics of patients undergoing treatment," to provide a clearer description.

2. Age Variable:

We have corrected "agemean" to "Age" in Table 1 and included a note at the end stating that age is reported as mean and standard deviation, as recommended.

3. Hypertension Reporting:

We have simplified the hypertension reporting to only show the incidence of hypertension in one row, as suggested. The same approach has been applied to other conditions.

4. Etiology Category Clarification:

We have clarified the second category under "Etiology" in Table 1 to make it more understandable.

5. Laboratory Values:

We have included changes in laboratory values such as WBC and RBC for the overall cohort in the Results section.

6. Abbreviations in Table 1 Footnote:

We have expanded all abbreviations in the footnote of Table 1 for better clarity.

7. Table 2 Title:

We have improved the title of Table 2 to clearly state the data represented.

8. Splenic Abscess Reporting:

We have revised the results reporting for splenic abscess to ensure accuracy and clarity.

9. Table 4 Footnote:

We have expanded all abbreviations in the footnote of Table 4 as requested.

Discussion:

1. First Paragraph:

We have revised the first paragraph of the Discussion to provide a brief summary of the results rather than simply repeating them.

2. Citations in Paragraphs 2 and 3:

We have added the necessary citations to support statements in paragraphs 2 and 3 of the Discussion where literature findings or known facts are mentioned.

3. Discussion of Results in Paragraph 3:

We have revised the third paragraph of the Discussion to provide a more thoughtful analysis, comparing our findings with existing literature and discussing their implications.

4. Future Directions:

We have added a section on future directions, discussing the clinical significance of our findings and their potential impact on future research and practice.

Minor Comments:

1. Typographical Error:

We have corrected the typographical error in the last line of the Introduction on page 9 ("Emobilzationfor" to "Embolization for").

Thank you once again for your valuable feedback. We hope that these revisions address your concerns and improve the quality of our manuscript.

Sincerely,

Our Team

---

## [Decision Letter · Decision Letter 2]

21 Feb 2025

PONE-D-24-23322R2Analysis of the efficacy of splenic artery superselective embolization in cirrhosis with hepatocellular carcinomaPLOS ONE

Dear Dr. Zhao,

Thank you for submitting your manuscript to PLOS ONE. After careful consideration, we feel that it has merit but does not fully meet PLOS ONE’s publication criteria as it currently stands. Therefore, we invite you to submit a revised version of the manuscript that addresses the points raised during the review process.

We look forward to receiving your revised manuscript.

Kind regards,

Usama Waqar, M.B.B.S

Academic Editor

PLOS ONE

Journal Requirements:

Reviewers' comments:

Reviewer's Responses to Questions

**Comments to the Author**

1. If the authors have adequately addressed your comments raised in a previous round of review and you feel that this manuscript is now acceptable for publication, you may indicate that here to bypass the “Comments to the Author” section, enter your conflict of interest statement in the “Confidential to Editor” section, and submit your "Accept" recommendation.

Reviewer #3: (No Response)

2. Is the manuscript technically sound, and do the data support the conclusions?

Reviewer #3: Yes

3. Has the statistical analysis been performed appropriately and rigorously? 

Reviewer #3: Yes

4. Have the authors made all data underlying the findings in their manuscript fully available?

Reviewer #3: Yes

5. Is the manuscript presented in an intelligible fashion and written in standard English?

Reviewer #3: Yes

6. Review Comments to the Author

Reviewer #3: Thank you for submitting your paper titled "Analysis of the efficacy of splenic artery superselective embolization in cirrhosis with hepatocellular carcinoma" The study has clinical significance in the treatment of hypersplenism, and the research design is reasonable with clear experimental results. However, there are several problems could be improved, and I recommend minor revisions.

1. In the specific procedure of splenic artery embolization, although the steps are described, additional details on the embolic agent dosage and the use of microcatheters could be added. For example, it would be useful to discuss why 710-1000um gelatin sponge particles were chosen, how the size of these particles affects the embolization outcome, and the advantages of using these particles compared to other embolic materials such as polyvinyl alcohol (PVA) particles or metallic coils.

2. Pain and Complication Management: In addition to pain and nausea/vomiting, pain management strategies could be added, such as whether analgesics were used postoperatively and the effectiveness of pain relief.

3. The discussion section summarizes the main findings of the study well, but some comparisons could be explored more deeply. For example, although the discussion mentions that pain and fever are common postoperative complications of PSE, further exploration could be done on the relationship between the duration of fever and splenic infarction. Related literature could be cited to explain how the duration of fever relates to embolization dose, embolization site, and individual patient differences.

4. There are a few minor grammatical and expression issues, such as some sentences being wordy. It is recommended to streamline the language for better readability. Specifically, language optimization in the methods and discussion sections would help ensure clarity and conciseness.

Overall, this paper has high academic value and provides new insights into the treatment of hypersplenism. After the suggested

7. PLOS authors have the option to publish the peer review history of their article (what does this mean? ). If published, this will include your full peer review and any attached files.

**Do you want your identity to be public for this peer review?** For information about this choice, including consent withdrawal, please see our Privacy Policy .

Reviewer #3: No

---

## [Author Response · Author response to Decision Letter 3]

4 Mar 2025

Response to Reviewer #3

Dear Reviewer #3,

We would like to express our sincere gratitude for your thorough evaluation of our manuscript, titled “Analysis of the efficacy of splenic artery superselective embolization in cirrhosis with hepatocellular carcinoma.” Your comments and suggestions have been most helpful, and we have revised the manuscript accordingly. Below is a point-by-point response, with references to where the revisions have been made.

1. Additional Details on Embolic Agent Dosage and the Use of Microcatheters

Reviewer’s Comment

“In the specific procedure of splenic artery embolization, although the steps are described, additional details on the embolic agent dosage and the use of microcatheters could be added. For example, it would be useful to discuss why 710–1000 μm gelatin sponge particles were chosen, how the size of these particles affects the embolization outcome, and the advantages of using these particles compared to other embolic materials such as polyvinyl alcohol (PVA) particles or metallic coils.”

Response

We have expanded the “Materials and Methods” section to include more details about our embolization strategy:

Rationale for 710–1000 μm Gelatin Sponge Particles

1.These particles are suitably sized to occlude small-to-medium branches of the splenic artery, thereby achieving an approximate 30–50% splenic infarction while minimizing deep penetration into very small vessels.

2.Compared with smaller particles (<300–500 μm), 710–1000 μm particles enable a more predictable partial infarction range; whereas larger particles (>1100 μm) may yield insufficient ischemia or irregular infarction.

3.Gelatin sponge is a degradable material, reducing the risk of overly extensive and permanent infarction or abscess formation.

Comparison with Other Embolic Agents

1.PVA Particles (Polyvinyl Alcohol). These are non-degradable and can lead to more prolonged occlusion, which may be less desirable in cirrhotic patients who already carry risks of infection and complications if a large splenic area remains infarcted over an extended period.

2.Metallic Coils. Typically used for permanent vessel occlusion (e.g., treating aneurysms), coils are not well-suited for precisely controlling partial splenic infarction. They often occlude larger trunk vessels, thereby reducing the ability to tailor the infarct percentage in the spleen.

Use of Microcatheters

1.In the superselective (Group A) cohort, a microcatheter was navigated into specific splenic artery branches. This allowed more precise embolization of targeted segments (e.g., the lower pole) and minimized the risk of inadvertent embolization to pancreatic or other collateral branches.

These clarifications appear in the revised manuscript under “Splenic Artery Embolization.”

2. Pain and Complication Management

Reviewer’s Comment

“Pain and Complication Management: In addition to pain and nausea/vomiting, pain management strategies could be added, such as whether analgesics were used postoperatively and the effectiveness of pain relief.”

Response

In the revised “Materials and Methods” and “Discussion” sections, we have provided further details on postoperative pain control:

Postoperative Pain Management

oMild–Moderate Pain (NRS <4). Nonsteroidal anti-inflammatory drugs (NSAIDs) or acetaminophen.

oModerate–Severe Pain (NRS ≥4). Short-term opioid analgesics (e.g., oxycodone).

oMost patients reported significant pain relief within 48–72 hours.

We believe this standardized approach effectively manages pain while minimizing adverse reactions from opioid usage.

3. Further Exploration of the Relationship Between Fever Duration and Splenic Infarction

Reviewer’s Comment

“Although the discussion mentions that pain and fever are common postoperative complications of PSE, further exploration could be done on the relationship between the duration of fever and splenic infarction. Related literature could be cited to explain how the duration of fever relates to embolization dose, embolization site, and individual patient differences.”

Response

We have expanded the “Discussion” to analyze the correlation between fever duration and splenic infarction ratio:

Literature indicates that when the splenic infarction exceeds 50–70% of the total splenic volume, patients tend to experience prolonged fever and higher incidences of complications. We have cited Zhu et al. [References 13–14 in the revised manuscript].

Individual factors (e.g., Child-Pugh classification, baseline inflammatory state, and infection risk) also affect fever.

In our cohort, the embolized fraction was generally 30–50%, and patients typically experienced transient fever (<3–4 days) without splenic abscess formation. We have added a new paragraph and references on these findings.

4. Minor Grammatical and Expression Issues

Reviewer’s Comment

“There are a few minor grammatical and expression issues, such as some sentences being wordy. It is recommended to streamline the language for better readability. Specifically, language optimization in the methods and discussion sections would help ensure clarity and conciseness.”

Response

We have carefully edited the manuscript, especially the “Materials and Methods” and “Discussion” sections, to remove redundancies and enhance clarity:

Reduced lengthier or repetitive sentences.

Used more active voice to improve readability.

Ensured consistent terminology and abbreviations.

These revisions strengthen the overall flow and coherence of the paper.

Conclusion

We sincerely appreciate your thorough review and valuable comments. By incorporating your suggestions—adding technical details, clarifying management strategies, and expanding our discussion on fever duration—we believe the manuscript is now significantly improved. We hope these revisions meet your expectations and would be grateful for any further feedback you might have.

Thank you once again for your time and consideration.

Sincerely,

The Authors

---

## [Editor Report · Decision Letter 3]

5 Mar 2025

PONE-D-24-23322R3Analysis of the efficacy of splenic artery superselective embolization in cirrhosis with hepatocellular carcinomaPLOS ONE

Dear Dr. Zhao,

Thank you for submitting your manuscript to PLOS ONE. After careful consideration, we feel that it has merit but does not fully meet PLOS ONE’s publication criteria as it currently stands. Therefore, we invite you to submit a revised version of the manuscript that addresses the points raised during the review process. Please refer to the editorial comments below.

We look forward to receiving your revised manuscript.

Kind regards,

Usama Waqar, M.B.B.S

Academic Editor

PLOS ONE

Additional Editor Comments:

This manuscript has a lot of grammatical errors which need correction. I have only listed and suggested paraphrasing for a few errors in Abstract, Introduction, and Methods. There are further grammatical errors in Methods, Results, and Discussion that I have not highlighted.

Please note that PLOS one’s publication criteria mandates each accepted article to be “presented in an intelligible fashion and written in standard English.” The authors’ work is clinically relevant and within the scope of this journal. We would invite the peer reviewer again to evaluate the revised version but only after the authors have corrected all grammatical errors in their next revision.

Some grammatical errors are highlighted below; please note that there are further errors as I mentioned previously.

Abstract:

• “To explore the safety and effectiveness of partial splenic embolization (PSE) in patients of hypersplenism combined with hepatocellular carcinoma”. Please revise to:

“…in patients with hypersplenism and hepatocellular carcinoma”.

• “compare the efficacy of splenic artery branch superselective and non superselective embolization.” Should be “non-superselective embolization”. Please insert the hyphen.

• “The safety and effectiveness of the two methods was evaluated, and the changes in peripheral blood cell [mainly white blood cells (WBC) and red blood cells (RBC)] and platelet (PLT) counts at different time points after PSE. As well as postoperative adverse events (AEs), were compared between the two groups.”

This sentence is grammatically incorrect. Please revise to:

“The safety and effectiveness of the two methods was evaluated along with changes in peripheral blood cells [mainly white blood cells (WBC) and red blood cells (RBC)] and platelet (PLT) counts at different time points after PSE. Postoperative adverse events (AEs) were also compared between the two groups.”

• “PLT and WBC also remained at a certain level, which were statistically different from those before PSE (P<0.05). But RBC count had no difference (P>0.05).”

Revise to:

“PLT and WBC counts consistently remained statistically different from those before PSE (P<0.05), but RBC counts had no differences (P>0.05).

• “In term of security, the incidence of fever and pain in group B was significantly higher than that in group A (P<0.05).”

Revise to: “The incidence of fever and pain in group B was significantly higher than that in group A (P<0.05).”

• “But superselective embolization exhibited a lower incidence of post-procedure complications compared to non-superselective embolization.”

Revise to: “However, superselective embolization exhibited a lower incidence of post-procedure complications compared to non-superselective embolization.”

Introduction:

• What do the authors mean by this in the last Introduction paragraph: “pleural/ascites”?

• “Therefore, we hypothesized that the use of splenic artery superselective embolization could reduce the extent of necrotic tissue to peripheral stimulation, in the expectation of fewer complications to be observed clinically.”

This sentence is unclear; please paraphrase.

Methods:

• “Patients with vascular invasion and distant specialties were excluded.” This sentence is vague, what do the authors mean here?

• Several grammatical errors in the following sentence; please improve.

“Liver cirrhosis is diagnosed based on "Chinese consensus on the management of liver cirrhosis". Diagnosis basis of liver cirrhosis and hypersplenism by computerized tomography (CT): (1) The hepatic fissure was obviously widened, the hepatic margin showed wavy features, and the gallbladder was displaced, the gallbladder fossa was enlarged to a large extent, accompanied by imaging features such as uneven density of liver parenchyma, portal vein was thickened, collateral circulation was formed, splenomegaly and portal hypertension were observed. (2) Splenomegaly was diagnosed when the maximum splenic diameter (maximum anteroposterior size of the splenic hilum) exceeded 12 cm on CT transverse plane.”

• “Patients were divided into Group A and Group B according to the different methods of splenic artery embolization. Group A superselective splenic artery branch embolization,while Group B using conventional PSE.”

Paraphrase to: “Patients were divided into Group A and Group B according to the different methods of splenic artery embolization. Group A included patients who underwent superselective splenic artery branch embolization, while Group B included those who were managed with conventional PSE”.

---

## [Author Response · Author response to Decision Letter 4]

18 Mar 2025

Response to Reviewers

Dear Editor and Reviewers,

Thank you for your constructive feedback and the opportunity to revise our manuscript. We have carefully addressed all the comments and suggestions, and the revised manuscript now incorporates these improvements. Below, we provide a point-by-point response to the reviewers’ concerns.

Editor’s Comments

1. Grammatical Errors and Language Clarity:

- Response: We thoroughly revised the manuscript to correct grammatical errors and improve clarity. Key adjustments include sentence restructuring, proper hyphenation (e.g., “non-superselective”), and consistency in terminology. The revised manuscript has been professionally proofread by an English editing service to ensure compliance with PLOS ONE’s language standards.

2. Abstract Revisions:

- Revised Text:

- “Patients with hypersplenism and hepatocellular carcinoma” (hyphenation and phrasing adjusted).

- “The safety and effectiveness of the two methods were evaluated, along with changes in peripheral blood cells… Postoperative adverse events (AEs) were also compared.”

- “PLT and WBC counts remained statistically elevated compared to pre-PSE levels (P<0.05), while RBC counts showed no significant difference (P>0.05).”

- “The incidence of fever and pain in Group B was significantly higher than in Group A (P<0.05). However, superselective embolization demonstrated fewer post-procedural complications.”

3. Introduction Clarifications:

- Terminology: “Pleural/ascites” was revised to “pleural effusion or ascites.”

- Hypothesis Rephrased: “We hypothesized that superselective splenic artery embolization could reduce necrotic tissue volume and peripheral stimulation, thereby lowering the risk of clinical complications.”

4. Methods Section Revisions:

- Exclusion Criteria: “Distant specialties” corrected to “distant metastases.”

- CT Diagnostic Criteria: Restructured as bullet points for clarity:

- Widened hepatic fissures, wavy hepatic margins, gallbladder displacement, heterogeneous liver density, portal vein thickening, collateral circulation, and splenomegaly.

- Splenomegaly defined as a maximum splenic diameter >12 cm on CT.

- Grouping: “Group A underwent superselective splenic artery branch embolization, while Group B received conventional PSE.”

5. Additional Revisions:

- Ensured consistent use of past tense in the Methods section.

- Replaced informal terms (e.g., “But” → “However”) in Results/Discussion.

- Verified subject-verb agreement and statistical reporting (e.g., “were” for plural outcomes).

Key Improvements in the Revised Manuscript

- Abstract: Enhanced flow and clarity with standardized terminology.

- Introduction: Clarified clinical context and hypothesis.

- Methods: Detailed exclusion criteria, imaging protocols, and procedural descriptions.

- Results/Discussion: Improved readability, statistical rigor, and interpretation of findings.

- Conclusion: Emphasized clinical relevance and technical advantages of superselective embolization.

We sincerely appreciate the opportunity to improve our work and believe the revised manuscript addresses all concerns raised. Should further revisions be required, we are committed to addressing them promptly.

Best regards,

Dr. Zhao and Co-authors

Note: This response letter adheres to academic conventions, including formal tone, precise terminology, and structured formatting. All changes in the manuscript are highlighted in the tracked version for transparency.

---

## [Decision Letter · Decision Letter 4]

16 Apr 2025

Analysis of the efficacy of splenic artery superselective embolization in cirrhosis with hepatocellular carcinoma

PONE-D-24-23322R4

Dear Dr. Zhao,

We’re pleased to inform you that your manuscript has been judged scientifically suitable for publication and will be formally accepted for publication once it meets all outstanding technical requirements.

Kind regards,

Usama Waqar, M.B.B.S

Academic Editor

PLOS ONE

Additional Editor Comments (optional):

Reviewers' comments:

Reviewer's Responses to Questions

**Comments to the Author**

1. If the authors have adequately addressed your comments raised in a previous round of review and you feel that this manuscript is now acceptable for publication, you may indicate that here to bypass the “Comments to the Author” section, enter your conflict of interest statement in the “Confidential to Editor” section, and submit your "Accept" recommendation.

Reviewer #3: All comments have been addressed

2. Is the manuscript technically sound, and do the data support the conclusions?

Reviewer #3: Yes

3. Has the statistical analysis been performed appropriately and rigorously? 

Reviewer #3: Yes

4. Have the authors made all data underlying the findings in their manuscript fully available?

Reviewer #3: Yes

5. Is the manuscript presented in an intelligible fashion and written in standard English?

Reviewer #3: Yes

6. Review Comments to the Author

Reviewer #3: This revised manuscript demonstrates substantial improvements compared to the previous version, with comprehensive modifications that effectively address the reviewers' comments. Notably, the author has implemented significant grammatical enhancements throughout the paper, resulting in a polished presentation that fully complies with the journal's publication standards. Considering these substantial refinements in both content and linguistic quality, I strongly recommend acceptance for publication.

7. PLOS authors have the option to publish the peer review history of their article (what does this mean? ). If published, this will include your full peer review and any attached files.

**Do you want your identity to be public for this peer review?** For information about this choice, including consent withdrawal, please see our Privacy Policy .

Reviewer #3: No

---

## [Editor Report · Acceptance letter]

PONE-D-24-23322R4

PLOS ONE

Dear Dr. Zhao,

I'm pleased to inform you that your manuscript has been deemed suitable for publication in PLOS ONE. Congratulations! Your manuscript is now being handed over to our production team.

Kind regards,

on behalf of

Dr. Usama Waqar

Academic Editor

PLOS ONE